# Dietary and Activity Factors Influence Poor Sleep and the Sleep-Obesity Nexus among Children

**DOI:** 10.3390/ijerph16101778

**Published:** 2019-05-20

**Authors:** Bridget Morrissey, Steven Allender, Claudia Strugnell

**Affiliations:** 1Global Obesity Centre, Centre for Population Health Research, Deakin University, Geelong 3220, Australia; steven.allender@deakin.edu.au (S.A.); claudia.strugnell@deakin.edu.au (C.S.); 2School of Health and Social Development, Deakin University, Geelong 3220, Australia

**Keywords:** sleep, children, physical activity, screen time, sugar sweetened beverage, weight status, obesity, sleep quality, sleep problem

## Abstract

*Background*: Behavioral factors such as physical activity, sedentary behavior and diet have previously been found to be key modifiable determinants of childhood overweight and obesity, yet require further investigation to provide an understanding of their potential influence on sleep outcomes along with the sleep-obesity nexus. *Methods*: The study included 2253 students (ages 8.8–13.5) from two monitoring studies across regional Victoria. Students completed a self-report electronic questionnaire on demographic characteristics, health behaviors (including sleep, physical activity, screen time and diet) and well-being, and were invited to have anthropometric measurements (height and weight) taken. Regression models were used to assess the associations between sleep, behavioral factors and BMI z-scores. *Results*: Screen time (particularly in bed) and sugar-sweetened beverage (SSB) consumption were shown to increase the likelihood of having more than three sleep problems, while physical activity and other dietary factors were not. After controlling for these behaviors, significance remained for having two or more than three sleep problems and an increased odds of overweight/obesity. *Conclusions*: This study highlights how the usage of screen devices and SSB consumption behaviors might influence children’s weight status via the sleep-obesity nexus.

## 1. Introduction

Getting a sufficient night’s sleep is widely acknowledged as an important restorative behavior and an important factor for a child’s growth, development, wellbeing and overall health [1,2,3]. An accumulating body of literature has recently taken to investigating the nexus between insufficient sleep and the current global epidemic of obesity among children [2,4]. Predominantly, this research has referred to insufficient sleep (sometimes referred to as sleep deprivation) in terms of duration, or amount, of time asleep deemed adequate for optimal outcomes [5]. This is in part due to guidelines that recommend approximately 9–11 h of sleep per night for children ages 5–12 [1,6]. Using this framing, representative data of children and adolescents in Canada and the USA report approximately 25% to one third are sleep deprived [7,8]. Another study of 9- to 11-year-olds across 12 different countries (Australia, Brazil, Canada, China, Colombia, Finland, India, Kenya, Portugal, South Africa, the United Kingdom and the United States) indicated that 58% of children slept less than the recommended guidelines [9]. While trends in Australia are less clear, a sample of 3495 children aged 5–15 years old in South Australia reported 24% had insufficient sleep [10].

The concept of a sleep-obesity nexus is supported by observed global trends in declining sleep durations among children occurring concurrently with detected increases in the prevalence of childhood overweight and obesity [2,4]. Research on this sleep-obesity nexus, amongst both cross sectional and longitudinal samples, has consistently demonstrated that shorter sleeps among children increase the risk of overweight/obesity [10,11,12,13,14,15,16]. A meta-analysis of 30,002 children from around the globe reported an increased odds of obesity (pooled OR = 1.89 95% CI = 1.43–1.68) amongst children with insufficient sleep durations compared with their longer sleeping counterparts [15].

Many authors are now indicating that duration of sleep alone is inadequate in determining the sufficiency of sleep [11,17,18]. They argue that sleep is multidimensional, and any consideration of sleep sufficiency should include elements such as the “quality” (perceived satisfaction, or structure of sleep wave cycles), “efficiency” (the ability to initiate and maintain sleep in an efficient manner); and “timing” (the placement of sleep within the 24 h of the day) [11,17,18]. These dimensions have been shown to play noteworthy roles in the sleep-obesity nexus, with poor quality, poor efficiency and later timings leading to poorer weight status among children, independent of sleep duration [5,18,19]. Considerably higher odds for obesity have been reported amongst children aged 9–16 years old with sleep-wake categories of late to bed/early to rise (OR = 1.55, 95% CI = 1.12–2.14) or late to bed/late to rise (OR = 1.47, 95% CI = 1.14–1.90), when compared with children with a sleep-wake category of early to bed/late to rise [20]. Jarrin et al. (2013) demonstrated that independent from adolescents’ sleep durations, participants’ risks of overweight and obesity were greater if they presented with poorer sleep quality, poor sleep maintenance/efficiency or delayed sleep timing [18]. Furthermore, Morrissey et al. [21] found that while scoring poorly on any one sleep dimension did not increase odds of obesity (OR = 1.17, 95% CI = 0.90–1.54), an increased odds of obesity was shown for having a combination of any two (OR = 1.68, 95% CI = 1.29–2.18), three (OR = 1.45, 95% CI = 1.03–2.01) or four or more (OR = 2.25, 95% CI = 1.41–3.58) sleep issues. These results suggest that the multiple sleep dimensions must be considered simultaneously in order to gain a stronger understanding of the sleep-obesity nexus.

Globally, the prevalence of sleep problems among children is of growing concern. Between 23% and 27% of children experience some type of sleep disorder [22,23,24,25]. A longitudinal study of Finnish children between 1989 and 2005 demonstrated that the prevalence of frequent sleep problems increased from 15% to 20% among girls [26]. While the drivers of this high prevalence are not fully understood, today’s profuse availability of food products (leading to higher caloric consumption) [27] as well as an exponential growth in technology (leading to higher daily use of screen devices) [28,29] are both suggested as potential influential factors [29,30]. Sleep problems have also displayed a bidirectional negative association with physical activity (PA) levels among 13-year-old girls in Estonia [31]. 

In terms of the sleep-obesity nexus, evidence of children with lower sleep quality engaging in less daytime PA [32] may in turn increase their risk of overweight and obesity. Furthermore, chronic sleep restriction for five consecutive nights has been shown to lead to increased consumption of high glycemic index foods among a sample of adolescents [33], contributing to energy imbalance and increased risk of obesity. So whilst sleep deprivation may not directly influence weight gain among children, it may considerably increase the risk of overweight and obesity as a result of associated behavioral determinants (e.g., physical activity, sedentary behavior and diet).

Behavioral factors such as physical activity, sedentary behavior and diet have previously been accepted as key modifiable determinants of childhood overweight and obesity, yet require further investigation to provide understanding of their potential influence on the sleep-obesity nexus [34,35]. This study aims to:investigate the influence of daily physical activity (PA), screen time (ST) and dietary behaviors on children’s sleep dimensions; andexamine whether the association between children’s sleep and weight status is modified by daily physical activity, screen time and dietary behaviors.

## 2. Materials and Methods 

### 2.1. Study Sample

The sample comprised data of Grade 4 (aged approx. 9–10 years) and Grade 6 (aged approx. 11–12 years) students participating in the Great South Coast Childhood Obesity Monitoring Study (GSC) [36] and the Goulburn Valley Health Behaviours Monitoring Study (GV) [37], collected between July 2016 and June 2017. Detailed information on the study design, sampling strategy, school and student recruitment process and measure have previously been published for the GSC study [36], which has been replicated in the GV study [37]. 

Briefly, all 146 primary schools (government, independent and catholic) within the nine local government areas of the Great South Coast region of Victoria (Colac-Otway, Corangamite, Southern Grampians, Glenelg, Moyne, Warrnambool) and Goulburn Valley region (Shepparton, Moira and Strathbogie) were invited to participate, of which 64% agreed (participation rates: GSC 56 of 83 schools; GV 38 of 63 schools). A passive (opt-out) consent approach was used to invite all Grade 4 and 6 students within each participating school (GSC N = 2197; GV N = 1356). Children were excluded from participation if their parent or guardian completed an opt-out form, or if the student verbally declined on the day (GSC N = 228; GV N = 300). Children were also free to participate in as little or as much of the data collection process as they wished (e.g., just have their height measured and not their weight, just complete the self-report behavioral questionnaire, etc.).

In total, data from 2862 Grade 4 and Grade 6 children (GSC N = 1817; GV N = 1045; combined participant response rate 85%) from 94 primary schools across Victoria were collected. All data collection took place at the school during school time. Students were invited to have anthropometric measurements taken and to complete a self-report electronic questionnaire that examined demographic characteristics, health behaviors and well-being.

This study received ethical approvals from Deakin University’s Human Research Ethics Committee (DUHREC 2014-279), the Victorian Department of Education and Training (DET 2015_002622) and the Catholic Archdiocese of Sandhurst and Ballarat.

### 2.2. Demographic Information

Children completed self-report items on date of birth and gender, as well as providing information on ancestry, primary language and socioeconomic position (via postcode). School Index of Community Socio-Educational Advantage (ICSEA) scores were also retrieved from the MySchool website [38]. Developed by Australian Curriculum, Assessment and Reporting Authority (ACARA) [39], ICSEA takes into account both student- and school-level factors to summarize educational advantages and disadvantages, with 1000 being the average benchmark score.

### 2.3. Sleep

A 16-item questionnaire was developed from previous studies and questionnaires [40,41,42,43,44,45,46,47,48,49,50,51] to assess aspects of children’s sleep (duration, quality, efficiency and timing) and behavioral factors. A detailed description including reliability and validity properties (where possible) of these are available (forthcoming).

Children self-reported their usual bedtimes and wake times on school nights within 15-min increments, allowing for the calculation of sleep duration (the difference between the two time points); with short sleep defined as less than nine hours, as per national guidelines [6,52]. Median splits of bed and wake times were used to determine categorization early/late bedtimes and wake times, following previously used criteria [20,41,53]. Here, late to bed was determined as 8:30 p.m. or later, and early to rise as 7:00 a.m. or earlier, similar to previous literature [41,53,54,55]. From this, sleep-wake categories were categorized as either: early to bed/late to rise (≤8:15 p.m./≥7:15 a.m.); early to bed/early to rise (≤8:15 p.m./≤7:00 a.m.); late to bed/late to rise (≥8:30 p.m./≥7:15 a.m.); or late to bed/early to rise (≥8:30 p.m./≤7:00 a.m.). Sleep quality was based on the questionnaire item “overall, how well do you think you sleep” with children required to select a response on a six-option Likert scale ranging from “Very Good” to “Very Bad”. These response options were dichotomized as “good” or “bad/very bad” sleep quality. Efficiency was assessed via questions on sleep initiation (“Over the last two weeks, have you found it hard to fall asleep (longer than 20 mins)”) and maintenance/waking episodes (“Some people wake up during the night, others never do. How often did you wake up”). Sleep initiation had a five-option Likert scale (ranging from “never” to “almost always”), and was dichotomized as poor from responses of “often” or “almost always”. Poor sleep maintenance/waking episode responses were determined as reporting having three or more waking episodes per night from a four-option Likert scale ranging from “never (I don’t wake up during the night)” to “often (3 or more times per night)”. 

### 2.4. Weight Status

Trained staff, using a standardized protocol and equipment (portable stadiometer: HM200P stadiometer, Charder Electronic Co. Ltd, (Taichung City, Taiwan, China) and electronic weight scale: A&D Precision Scale UC-321, A&D Australasia Pty Ltd, (Adelaide, Australia), took measurements of height and weight. Measures were taken without shoes and wearing only light clothing. Each measure was taken twice (to the nearest 0.1 cm for height, and the 0.1 kg for weight), with a third measure taken if the difference was greater than 0.1 kg for weight or 0.5 cm for height. The average of two or three measurements was used to generate each child’s height and weight. Participants’ height, weight, gender and age were used to calculate BMI (Body Mass Index) z-score and weight status, using the WHO growth standards for children ages 5–19 [56].

### 2.5. Physical Activity, Screen Time and Dietary Behaviors 

Based on items from the Core Indicators and Measures of Youth Health survey [57], students were asked to select how much time they had been physically active for each of the last seven days, with six possible duration options (“none”, “1 to 14 min”, “15 to 29 min”, “30 to 59 min”, “1 to 2 h” or “more than two hours”). Students then reported screen usage (outside of school hours) for each of the last seven days, selecting from five duration options (“none”, “less than 1 hour”, “1 to 2 h”, “more than 2 h, but less than three” or “more than 5 h”). Items adapted from the Children’s Sleep Hygiene and the Adolescent’s Sleep Hygiene questionnaires [58,59] assessed physical activity and screen-use behaviors around bedtime over the last two weeks. Children were asked whether they had “been very active (e.g., playing sports, playing outside, running, wrestling)” or “used electronic devices (e.g., computer/gaming console/ tablet/phone)” in the one hour prior to bedtime, with responses ranging from “never” to “almost always”. The last question was then repeated to assess usage of these devices while in bed/during bedtime.

Consumption of fruit and vegetables, sugar-sweetened beverages (SSBs), snack foods and takeaway (meals purchased from restaurants or fast-food vendors) were assessed from items extracted from the Simple Dietary Questionnaire [60,61]. Students selected from options starting from “none/don’t eat” to “7 or more per day” (options increasing by half servings) for both average fruit and vegetable consumption. Options ranging from “rarely or never” to “almost every day” and up to “three times per day” (eight options total) were used to report consumption of snack foods and SSBs. Takeaway food consumption options ranged from “rarely or never” to “2–4 times per week”, up to “every meal” (eight options total).

### 2.6. Data Management

Of the 2862 consenting students, 2253 were included in the analysis (816 GV; 1437 GSC); 169 students were excluded due to missing BMI z-scores, six due to extreme BMI z-scores being ±3 standard deviations from the mean and 434 due to inaccurate/missing data or irregular bed/wake times. Bedtimes and wake times were screened, with exclusion criteria representing inaccurate or irregular bed/wake times generated as described below. Cut-points for bedtime and wake times were created to exclude irregular and inaccurate/missing data. We excluded bedtimes reported before 6:00 p.m. or after 2:15 a.m. and wake times before 3:00 a.m. or after 8:45 a.m.

Results from each sleep variable were dichotomized as positive or negative (as outlined above), with negative defined as: sleeping less than nine hours per night, going to bed later than 8:30 p.m.; waking earlier then 7:00 a.m.; sleep quality perceived as bad/very bad; an initiation problem of 20 min or more to fall asleep; and three or more waking episodes per night. From these, a sleep score was created to indicate the number of sleep dimensions scored as negative, out of a maximum of six. The sleep score was categorized ranging from 0 "no sleep problem" to 3 "three or more sleep problems".

Physical activity (PA), screen time (ST) and dietary behaviors were dichotomized to reflect whether the guidelines or recommendations were met, or to indicate a negative versus positive behavior. Based on the recommended one hour of moderate-to-vigorous PA per day and less than two hours of ST per day for leisure [62], children were categorized as meeting the guidelines if they reported being active for one or more hours, or less than two hours, of ST on five out of the seven days. Fruit guidelines were considered met if students reported consuming two servings of fruit per day. Vegetable guidelines were considered met if they reported consuming the recommended 5.5 servings per day for boys aged 12 years or older, or five servings per day for girls and younger boys [63]. Screen time, physical activity and sugar-sweetened beverage consumption before bed, along with screen usage in bed, were positively scored if reported to be never or almost never. Snack and average sugar-sweetened beverages consumption were positively scored if reported as being consumed once a day or less, and takeaway consumption as once a week or less. A detailed outline of each variable and the coding is available (Appendix A).

### 2.7. Statistical Analysis

All statistical analyses were conducted using STATA SE15 (Stata Corporation, College Station, TX, USA). Initial *t*-tests and chi-square tests were conducted to compare the percentages and proportions of main variables for each study sample (GV and GSC) separately (data not shown). As no significant difference for BMI z-score, weight status categorization, gender, proportion of English-only speakers, proportion per Grades 4 and 6 or the proportion categorized as short sleepers (<9 h per night), the samples were pooled for all further analyses. Bivariate chi-squared analysis was conducted to assess for associations between the dichotomized behavioral factors (including PA, ST and dietary variables) with weight status categories, and then with categories of poorly scored sleep dimensions (none/one sleep problem, two sleep problems, three or more sleep problems). Multivariate analysis was then conducted using logistical regressions to examine the odds of poor sleep scores depending on the dichotomous categorization of each behavioral variable (i.e., guidelines met or not met), adjusting for age, gender, ICSEA, study sample, BMI z-score and school clustering. Finally, multiple logistic regression models were conducted to assess the association between number of sleep problems and obesity. The initial analysis controlled study sample, age, gender, ICSEA and school clustering. The adjusted model additionally controlled for behavioral factors significantly associated with either sleep problems or weight status (data not shown) (PA guidelines, PA before bed, ST guidelines, ST before bed, ST in bed, sugar-sweetened beverage average consumption and before bed consumption).

## 3. Results

In this sample, 35.2% of children were classified as overweight or obese, 70% were drawn from the two most deprived SEIFA quintiles (Socio-Economic Indexes for Areas, according to schools) and 90.7% reported speaking English only at home (Table 1). No significant differences in proportion of overweight and obesity were reported for age or gender. When examining sleep dimensions by weight status, higher proportions of children were categorized as overweight/obese for those who: slept <9 h (42% <9 h vs 32% ≥11 h, *p* = 0.029); had later sleep patterns (41% late to bed/early to rise sleepers, compared with 31% early to bed/late to rise sleepers, *p* = 0.003); poor initiation (40% poor sleep initiation vs 33% good, *p* = 0.045); and poor quality (44% poor quality vs 34% good quality, *p* = 0.009). Higher proportions of overweight/obesity were found among those with a greater number of overall sleep problems (41% with three or more problems, compared with 32% with none or less, *p* < 0.001).

### 3.1. Physical Activity and Screen Time Behaviors

Initial chi-squared analysis (Table 2) indicated significant differences in screen time (ST) behaviors across the number of sleep problems, while no significant association was found for the physical activity (PA) behaviors. A lower proportion of children meeting the ST guidelines (<2 h ST per day) on five or more days (13%) were categorized with having three or more sleep problems (*p* < 0.001) compared with those not meeting the guidelines (21%). A higher proportion (23%) of those who engaged with screen devices within one hour prior to bedtime and those who used screen devices in bed (29%) had three or more problems, compared with those who did not use devices before or during bedtime and had three or more sleep problems (11% and 13%, respectively) (*p* < 0.001). These associations remained in the adjusted logistical regression (Table 3). Children who met the ST guidelines had lower odds of recoding two (OR = 0.78, CI = 0.64–0.94) or three or more sleep problems (OR = 0.55, CI = 0.42–0.73) than those who did not meet the guidelines. Additionally frequent use (often/always) of a screen device in the hour before going to bed, as well as usage whilst in bed, demonstrated higher odds of having two (OR = 1.61, CI = 1.30–2.00; and OR = 1.91, CI = 1.42–2.58, respectively) or three or more sleep problems (OR = 2.59, CI = 1.98–3.38; and OR = 3.26, CI = 2.27–4.67, respectively), compared with those who only sometimes/never used screen devices at these times. 

### 3.2. Dietary Behaviors

Of the dietary factors, initial chi-squared analysis indicated all consumption behaviors, except meeting the fruit and vegetable guidelines, differed across the number of sleep problems. The proportion of children with three or more sleep problems was higher amongst participants consuming two or more sugar-sweetened beverages (SSB) per day, or those consuming SSBs in the hour prior to bedtime (*p* < 0.01) (Table 2). This was supported in the multivariate analysis, where consuming two or more SSBs per day, or consuming an SSB in the hour prior to bedtime, increased the odds of having two sleep problems (OR = 1.72, CI = 1.19–2.43; and OR = 1.82, CI = 1.18–2.80, respectively) or three or more problems (OR = 1.96, CI = 1.37–2.80; and OR = 2.21, CI = 1.32–3.72, respectively), compared with consuming an SSB once per day and sometimes/never before bed (Table 3).

Takeaway and snacking consumption also indicated higher proportions of children with three or more sleep problems amongst those consuming takeaway twice or more per week (compared with once or less) (*p* < 0.05) and amongst those consuming snack foods twice or more per day (compared with once or less) (*p* < 0.01). However, adjusted regression analysis only found these behaviors to be associated with an increased odds of having two sleep problems (OR = 1.48, CI = 1.09–2.02; and OR = 1.42, CI = 1.15–1.77, respectively), but not three or more (OR = 1.24, CI = 0.87–1.77; and OR = 1.31, CI = 0.99–1.72, respectively) (Table 3). 

### 3.3. Impact on Weight Status

Logistic regressions of the relationship between overweight/obesity and children’s sleep and behavioral factors demonstrated increased odds of overweight or obesity among children with two sleep problems (OR = 1.48, CI = 1.24–1.77) and those with three or more sleep problems (OR = 1.48, CI = 1.11–1.97) compared with those with one or less sleep problems, after adjusting for age, gender, ICSEA, study sample and school clustering (Table 4, model 1). These findings were consistent in the final adjusted model (model 2), which additionally controlled for behavioral factors found to be associated with either sleep problem score or weight status (PA guidelines, PA before bed, ST guidelines, ST before bed, ST in bed, sugar-sweetened beverages average consumption and before bed consumption). This model confirmed that, compared with having one or no sleep problems, children with two sleep problems and those with three or more sleep problems had higher odds of being categorized as overweight or obese (OR = 1.42, CI = 1.18–1.72; and OR = 1.38, CI = 1.02–1.88, respectively).

Physical activity was the only behavioral factor associated with weight status in the adjusted model. Overweight/obesity categorization decreased amongst those who met the PA guidelines on five or more days (OR = 0.72, CI = 0.60–0.86) and for those being physically active 1 h before bedtime (OR = 0.64, CI = 0.50–0.81).

## 4. Discussion

This study explored behavioral aspects of physical activity, screen time and dietary factors and their association with poor sleep among children. Of the different behavioral components explored, only those involving the usage of screen devices and SSB consumption were shown to be associated with increased odds of having more than three sleep problems. Factors surrounding PA levels (in terms of either meeting the recommended guidelines, or being active in the 60 min directly before bedtime), or meeting guidelines for recommended fruit and vegetable consumption, did not influence the odds of reporting more sleep problems. While consumption of snack foods and takeaway increased the odds of having two sleep problems, this was not the case for having three or more sleep problems. Furthermore, converse to previous findings, overweight/obesity risk was not associated with ST and SSB consumption, with the current findings suggesting these behaviors might influence weight status via the sleep-obesity nexus.

The impact of TV and usage of electronic entertainment/communication devices (i.e., computers and smart phones) on insufficient sleep has previously been observed, mostly linking ST with reduced sleep durations among children [44,64,65,66]. Night-time access to and usage of electronic screen devices have been shown to lead to reductions in sleep durations among Grade 5 (ages 10–11) children in Canada [64]. Additionally, a study in the Netherlands reported children (ages 9–13) had reduced sleep durations when spending higher amounts of time viewing TV or using computers [66]. With literature currently limited, few studies have reported on the impact of ST on children’s dimensions of sleep (beyond sleep duration); however ST has been shown to impact other aspects of adults’ sleep efficiency, namely, increased trouble in falling asleep and more frequent waking during the night [67]. The current study builds on this understanding, noting that screen device behaviors are not only potentially detrimental to children’s sleep durations and sleep efficiency, but increase the risk of having issues across multiple dimensions of sleep. Those who reported engaging in ST during the hour before going to bed, as well as those using a screen while in bed, were approximately three times more likely to have three or more sleep problems. Conversely, children who mostly met the guidelines of less than two hours ST per day were 22% less likely to have two sleep problems, and 45% less likely to have three or more sleep problems (OR = 0.78, CI = 0.64–0.94; and OR = 0.55, CI = 0.42–0.73, respectively). Screen time appears to be a potentially influential factor in the sleep-obesity nexus, either through weight status outcomes determined directly via poor sleep, or via other potentially co-related obesogenic factors related to high ST such as higher sedentary time, decreased physical activity or negative dietary behaviors. Future initiatives for obesity among children should consider the impact ST has on sleep, along with these correlated behaviors.

Of the dietary factors assessed, the current study initially found those children with more sleep problems consumed more takeaway meals per week, more snacks per day and more SSB consumption, both daily and/or before bedtime. However, after control for demographic factors, only SSB consumption remained significant. 

Previous studies have shown the association between poor dietary behaviors and poor sleep. It is suggested that shorter sleep durations among children and adolescents might lead to an energy deficit, and therefore lead to higher consumption of energy/caloric dense foods in response [33]. Short sleepers have been shown to have higher consumption of candy [66], takeaway meals [44,66] and SSB [44], and conversely longer sleep durations have been associated with higher diet quality scores among children (β = 0.60, 95% CI = 0.11–1.09) [68].

It has also been suggested that diet quality might be linked with sleep timing, more so than sleep duration [69]. Golley et al. (2013) indicate that while sleep did not influence total energy consumption of children ages 9–16, compliance with the Dietary Guidelines Index for Children and Adolescents was impacted by sleep timing, with reduced compliance amongst later sleepers [69]. A late sleep preference among children and adolescents has been associated with increased consumption of takeaway food [70] and a higher SSB consumption [43]. While the current study did not support these previously reported associations between takeaway and snack consumption with sleep, this could be due to sample size and the small numbers of participants engaging in these behaviors with three or more sleep problems, reducing the power to detect an association.

The potential mechanisms behind these poor dietary behaviors and later sleep timings are suggested to be related to the extra time available for consumption, or consumption to provide an energy boost, which enables the delayed bedtime (i.e., consumption is due to sleep behaviors) [47]. However, a randomized crossover trial has demonstrated that sugar intake among a group of adults led to significantly impaired sleep with more waking episodes and a less restorative sleep [71]. The current study indicated that those consuming SSB before bed were more than twice as likely to have three or more sleep problems (OR = 2.21, CI = 1.32–3.72). This could indicate a relationship that may work in either direction, supporting both arguments above and linking poor sleep timing, duration, quality and efficacy with dietary behaviors. For example, those with poorer sleep habits, such as going to bed later, may be more likely to consume calorie-dense foods during this time, which in turn may affect their sleep quality and efficiency and lead to an energy deficit that further encourages the consumption of higher caloric dense foods the following day. Further studies would benefit from utilizing more representative monitoring samples to assess dietary behaviors and changes in sleep outcomes longitudinally, in order to improve analysis power and to determine the direction of this association among children. 

Contrary to previous reports, sleep was not shown to be associated with physical activity among the current sample. Previous studies have demonstrated that children with lower sleep efficiency [32] and shorter sleep durations [72] engage in less daytime PA. However, converse findings have also been reported, where higher PA levels were associated with lower sleep durations [65] and lower sleep efficiency among children [73]. The inverse association of sleep duration and PA reported by Wells et al. (2008) could be due to an error in the self-report measures for both behaviors [65], whereas Stone et al. (2013) [72] and Gupta et al. (2002) [32] used accelerometry to measure PA and sleep. Furthermore, while McNeil et al. (2014) [73] also objectively measured sleep via accelerometry, they noted an inverse association between PA and children’s sleep efficiency, which might be due to the difference in wrist [32] versus hip placement of devices [73], and 24-hour [32] versus seven-day monitoring [73]. Hip-worn accelerometry has been shown to have a tendency to overestimate sleep duration and sleep efficiency [74], and wear-times less than five days are shown to be less reliable than five or more days [75].

It has also been argued that, rather than overall PA engagement, factors of the intensity (relative to the individual’s fitness levels) and timing (both in terms of duration and placement in the day) could be imperative to understanding the impact PA might have on sleep behavior [76]. While the current study found no association between sleep score and either meeting the physical activity guidelines or being physically active before going to bed, future studies should potentially consider assessing these factors using reliably objective measures (such as accelerometry).

When assessing the sleep-obesity nexus, the current analysis found only physical activity factors (meeting the guidelines and physical activity before bed), and having more sleep problems were associated with children’s weight statuses. Supporting the consensus in the literature [77], decreased likelihood of overweight and obesity was found among those with greater physical activity levels, either by meeting the physical activity guidelines on at least five of seven days (OR = 0.72, CI = 0.60–0.86) or through physical activity prior to bedtime (OR = 0.64, CI = 0.50–0.81). More interesting, this study supports findings of previous literature [5] for a sleep-obesity nexus, even after controlling for other modifiable behaviors. Higher odds of overweight or obesity were found for children reporting any two (OR = 1.42, CI = 1.18–1.72) or three or more sleep problems (OR = 1.38, CI = 1.02–1.88) independent of PA, ST and dietary behaviors.

This further enhances the understanding of how children’s behavioral factors influence sleep and the sleep-obesity nexus. While usage of screen devices and dietary behaviors were not directly associated with weight status among the current sample, they were highlighted as potentially important modifiable behavioral factors influencing sleep. The association between SSB consumption and ST behaviors with sleep suggests these behaviors might influence weight status via the sleep-obesity nexus. These findings support the importance of adhering to the 24-hour movement guidelines outlined by the Australian Government Department of Health, considering the role of multiple behaviors on the healthy development and obesity risk among children and adolescents [78]. However, with current results suggesting the impact of sugar-sweetened beverage consumption on children’s sleep behaviors and weight status outcomes, there is a need for more specific guidelines on these behaviors. Current Australian guidelines recommend the intake of SSBs be limited [63]; however, it is recommended the Government to work with researchers to provide quantifiable guidelines to limit this behavior (similar to the recommended 2-h screen-time limit).

It is important to note the strengths and limitations of the current study. As mentioned, the current study benefitted from a large representative sample of children within two rural regions of Victoria, with high school (GSC = 68%, GV = 60%) and student (GSC = 90%, GV = 78%) response rates. However, although the current population groups did not differ significantly, it cannot be assumed that the reported associations can be generalized beyond children ages 8–13 within rural Victoria, Australia. Secondly, causal associations cannot be inferred, due to the cross-sectional data utilized in the current study. Furthermore, while questionnaire items were sourced from studies with previous validated question items [48,49,50] and similar self-report questionnaires validated among young children, the self-report nature of the sleep and behavioral-factor items could be subject to recall or report bias [79]. While there is evidence supporting the suitability of questionnaires in the examination of associations with weight status [80], objective measures (such as accelerometry) are deemed more accurate for the assessment of sleep and physical activity behaviors.

Future studies would benefit from the incorporation of more objective measures, such as accelerometry, to objectively assess behavioral factors of sleep and physical activity. They would also benefit from monitoring data from broader samples (beyond the two regions of Victoria), allowing the longitudinal assessment of these associations, which would be more generalizable beyond the current sample.

## 5. Conclusions

The current study suggests that behavioral factors, including usage of screen devices and the consumption of sugar-sweetened beverages, are influential factors on the sleep-obesity nexus. While dietary factors and screen-usage behaviors were not associated with children’s weight status, more frequent screen usage and the consumption of sugar-sweetened beverages were reported amongst those with more sleep problems. Furthermore, higher numbers of sleep problems increased overweight or obesity odds, regardless of controlling for all other behavioral factors. The outlined association between the number of sleep problems with the usage of screen devices and consumption of sugar-sweetened beverages could suggest that these factors might influence children’s weight statuses via the sleep-obesity nexus. Further developing this understanding through monitoring cohort studies could provide insight into strategies to improve children’s sleep and reduce overweight and obesity rates.

## Figures and Tables

**Table 1 ijerph-16-01778-t001:** Participant demographic characteristics and behaviors by weight status.

Participant Characteristics and Behaviors	All ^1^	Healthy Weight	Overweight/Obese	*p* Value ^2^
Participants, *n* (%)	2253	1461 (64.8)	792 (35.2)	
Age, mean (SD)	10.91 (1.1)	10.93 (1.1)	10.86 (1.1)	*p* = 0.147
School year, *n* (%)				
Year 4	1167 (51.8)	757 (64.9)	410 (35.1)	*p* = 0.983
Year 6	1086 (48.2)	704 (64.8)	382 (25.2)	
Sex, *n* (%)				
Male	1130 (50.2)	743 (65.8)	387 (34.2)	*p* = 0.367
Female	1123 (49.8)	718 (63.9)	405 (36.1)	
BMI z-score, mean (SD)	0.64 (1.1)	–0.04 (0.7)	1.89 (0.6)	
Language at home, *n* (%)				
English	2023 (90.7)	1315 (65.0)	708 (35.0)	*p* = 0.257
Other	208 (9.3)	127 (61.1)	81 (38.9)	
ICSEA				
Mean (±SD)	**995.5 (±51.9)**	**999.08(±51.3)**	**988.89(±52.5)**	***p* = 0.009**
SIEFA, *n* (%)				
Quintile 1 (lowest)	279 (19.6)	178 (63.8)	101 (36.2)	*p* = 0.762
Quintile 2	715 (50.2)	478 (66.9)	237 (33.2)	
Quintile 3	282 (19.8)	179 (63.5)	103 (36.5)	
Quintile 4	134 (9.4)	88 (65.7)	46 (34.3)	
Quintile 5	Suppressed	Suppressed	Suppressed	
Sleep duration, *n* (%)				
<9 h	**125 (5.6)**	**72 (57.6)**	**53 (42.4)**	***p* = 0.029**
9–11 h	**1379 (61.2)**	**879 (63.7)**	**500 (36.3)**	
>11 h	**749 (33.2)**	**510 (68.1)**	**239 (31.9)**	
Sleep-wake categories, *n* (%)				
EL	**467 (20.7)**	**323 (69.2)**	**144 (30.8)**	***p* = 0.003**
EE	**256 (11.4)**	**180 (70.3)**	**76 (29.7)**	
LL	**1120 (49.7)**	**717 (64.0)**	**403 (36.0)**	
LE	**410 (18.2)**	**241 (58.8)**	**169 (41.2)**	
Sleep initiation, *n* (%) ^3^				
Good	**1076 (48.5)**	**720 (66.9)**	**356 (33.1)**	***p* = 0.045**
Moderate	**499 (22.5)**	**328 (65.7)**	**171 (34.3)**	
Bad/very bad	**645 (29.8)**	**394 (61.1)**	**251 (38.9)**	
Perceived quality, *n* (%)				
Good	**1739 (77.6)**	**1153 (66.3)**	**586 (33.7)**	***p* = 0.009**
Moderate	**318 (14.2)**	**194 (61.0)**	**124 (39.0)**	
Bad/very bad	**183 (8.2)**	**103 (56.3)**	**80 (43.7)**	
Poor sleep score, *n* (%)				
≤1 sleep problems	**1350 (59.9)**	**925 (68.5)**	**425 (31.5)**	***p* < 0.001**
2 sleep problems	**566 (25.1)**	**337 (59.5)**	**229 (40.5)**	
≥3 sleep problems	**337 (15.0)**	**199 (59.1)**	**138 (40.9)**	
Physical activity guidelines, *n* (%) ^4^				
Met on 5 out of 7 days	**865 (38.4)**	**615 (71.1)**	**250 (28.9)**	***p* < 0.001**
Not met on 5 out of 7 days	1388 (61.6)	846 (61.0)	542 (39.0)	
Physically active 1 h before bed, *n* (%)				
Often/almost always	**1397 (63.0)**	**583 (71.0)**	**238 (29.0)**	***p* < 0.001**
Never/sometimes	**821 (37.0)**	**856 (61.3)**	**541 (38.7)**	
Screen time guidelines, *n* (%) ^5^				
Met on 5 out of 7 days	**1728 (76.7)**	**1,145 (66.3)**	**583 (33.7)**	***p* = 0.011**
Not met on 5 out of 7 days	**525 (23.3)**	**316 (60.2)**	**209 (39.8)**	
Screen time 1 h before bed, *n* (%)				
Never/sometimes	**1462 (65.6)**	**976 (66.8)**	**486 (33.2)**	***p* = 0.009**
Often/almost always	**766 (34.4)**	**469 (61.2)**	**297 (38.8)**	
Screen time in bed, *n* (%)				
Never/sometimes	1944 (87.3)	1273 (65.5)	671 (34.5)	*p* = 0.058
Often/almost always	283 (12.7)	169 (59.7)	114 (40.3)	
Fruit consumption, *n* (%) ^6^				
Guidelines not met	**532 (24.2)**	**322 (60.5)**	**210 (39.5)**	***p* = 0.019**
Guidelines met	**1666 (75.8)**	**1101 (66.1)**	**565 (33.9)**	
Vegetable consumption, *n* (%) ^7^				
Guidelines not met	**1873 (83.3)**	**1,199 (64.0)**	**674 (36.0)**	***p* = 0.045**
Guidelines met	**376 (16.72)**	**261 (69.4)**	**115 (30.6)**	
Sugar sweetened beverage consumption, *n* (%)				
Once a day or less	**2059 (91.7)**	**1350 (65.6)**	**709 (34.4)**	***p* = 0.040**
Twice or more per day	**186 (8.3)**	**108 (58.1)**	**78 (41.9)**	
Sugar sweetened beverage 1 h before bed, *n* (%)				
Never/sometimes	2105 (94.5)	1365 (64.9)	740 (35.1)	*p* = 0.747
Often/almost always	123 (5.5)	78 (63.4)	45 (36.6)	
Takeaway consumption, *n* (%)				
Once a week or less	**2012 (89.5)**	**1323 (65.8)**	**689 (34.2)**	***p* = 0.007**
Twice or more per week	**237 (10.5)**	**135 (57.0)**	**102 (43.0)**	
Snack consumption, *n* (%)				
Once a day or less	1563 (73.8)	1022 (65.4)	541 (34.6)	*p* = 0.945
Twice or more per day	555 (26.2)	362 (65.2)	193 (34.8)	

^1^ Percentages relate to proportion of sample overall; ^2^
*p*-value relates to significant difference between normal weight and overweight/obesity; ^3^ good: <20 min, moderate: sometimes >20 min, bad/very bad: mostly >20 min; ^4^ PA guidelines met ≥1 h/day on five or more days; ^5^ ST guidelines met ≤2 h/day on five or more days; ^6^ consumed ≥2 servings of fruit per day; ^7^ consumed ≥5.5 servings of vegetables for boys aged ≥12 years, or ≥5.5 servings for younger boys and all girls. BMIz: body mass index (age and sex adjusted); ICSEA: Index of Community Socio-Educational Advantage; SEIFA: Socio-Economic Indexes for Areas (according to schools); EL: early to bed/late to rise; EE: early to bed/late to rise; LL: late to bed/late to rise; LE: late to bed/late to rise; suppressed: values not shown due to very small participant numbers. Bold numbers indicate *p*-values <0.05 or <0.001.

**Table 2 ijerph-16-01778-t002:** Health behavioral characteristics by sleep score.

Health Behaviors	Number of Sleep Dimensions Scored as Poor			
	One or Less N = 1350	Two N = 566	Three or More N = 377	*p* Value ^1^
Physical activity guidelines, *n* (%) ^2^				
Met on 5 out of 7 days	543 (62.8)	205 (23.7)	117 (13.5)	*p* = 0.084
Not met on 5 out of 7 days	807 (58.1)	361 (26.0)	220 (15.9)	
Physically active 1 h before bed, *n* (%)				
Often/almost always	472 (57.5)	211 (25.7)	138 (16.8)	*p* = 0.117
Never/sometimes	857 (61.4)	345 (24.7)	195 (14.0)	
Screen time guidelines, *n* (%) ^3^				
Met on 5 out of 7 days	**1079 (62.4)**	**423 (24.5)**	**226 (13.1)**	***p* < 0.001**
Not met on 5 out of 7 days	**271 (51.6)**	**143 (27.2)**	**111 (21.1)**	
Screen time 1 h before bed, *n* (%)				
Never/sometimes	**962 (65.8)**	**339 (23.2)**	**161 (11.0)**	***p* < 0.001**
Often/almost always	**372 (48.6)**	**221 (28.9)**	**173 (22.6)**	
Screen time in bed, *n* (%)				
Never/sometimes	**1218 (62.7)**	**473 (24.3)**	**253 (13.0)**	***p* < 0.001**
Often/almost always	**113 (39.9)**	**88 (31.1)**	**82 (29.0)**	
Fruit consumption, *n* (%) ^4^				
Guidelines not met	299 (56.2)	142 (26.7)	91 (17.1)	*p* = 0.090
Guidelines met	1021 (61.3)	409 (24.6)	236 (14.2)	
Vegetable consumption, *n* (%) ^5^				
Guidelines not met	1138 (60.8)	456 (24.4)	279 (14.9)	*p* = 0.126
Guidelines met	211 (56.1)	110 (29.3)	55 (14.6)	
Sugar sweetened beverage consumption, *n* (%)				
Once a day or less	**967 (61.9)**	**368 (23.5)**	**228 (14.6)**	***p* = 0.003**
Twice or more per day	**298 (53.7)**	**164 (29.6)**	**93 (16.8)**	
Sugar sweetened beverage 1 h before bed, *n* (%)				
Never/sometimes	**1279 (60.8)**	**520 (24.7)**	**306 (14.5)**	***p* < 0.001**
Often/almost always	**53 (43.1)**	**40 (32.5)**	**30 (24.4)**	
Takeaway consumption, *n* (%)				
Once a week or less	**1226 (60.9)**	**491 (24.4)**	**295 (14.7)**	***p* = 0.023**
Twice or more per week	**123 (51.9)**	**74 (31.2)**	**40 (16.9)**	
Snack consumption, *n* (%)				
Once a day or less	**967 (61.9)**	**368 (23.5)**	**228 (14.6)**	***p* = 0.003**
Twice or more per day	**298 (53.7)**	**164 (29.6)**	**93 (16.8)**	

^1^ P-value relates to chi-square tests across sleep score; ^2^ PA guidelines met ≥1 h/day on five or more days; ^3^ ST guidelines met ≤2 h/day on five or more days; ^4^ consumed ≥2 servings of fruit per day; ^5^ consumed ≥5.5 servings of vegetables for boys aged ≥12 years, or ≥5.5 servings for younger boys and all girls. Bold numbers indicate *p*-values <0.05 or <0.001.

**Table 3 ijerph-16-01778-t003:** Unadjusted and adjusted logistic regression with odds ratio (OR) of behavioral factors by sleep score ^#^.

Health Behaviors	Unadjusted			Adjusted for Demographics ^1^		
	OR	*p*	95% CI	OR	*p*	95% CI
Physical activity guidelines ^2^ (0 = not met, 1 = met)						
≤1 sleep problem	1.00			1.00		
2 sleep problems	0.84	*p* = 0.079	0.70–1.01	0.83	*p* = 0.057	0.69–1.01
≥3 sleep problems	**0.79**	***p* = 0.047**	**0.63–1.00**	0.79	*p* = 0.052	0.62–1.00
Physically active 1 h before bed (0 = never/sometimes, 1 = often/always)						
≤1 sleep problem	1.00			1.00		
2 sleep problems	1.11	*p* = 0.337	0.90–1.38	1.13	*p* = 0.286	0.90–1.40
≥3 sleep problems	1.28	*p* = 0.060	0.99–1.67	1.30	*p* = 0.050	1.01–1.69
Screen time guidelines ^3^ (0 = not met, 1 = met)						
≤1 sleep problem	1.00			1.00		
2 sleep problems	**0.74**	***p* = 0.002**	**0.61–0.90**	**0.78**	***p* = 0.010**	**0.64–0.94**
≥3 sleep problems	**0.51**	***p* < 0.001**	**0.39–0.67**	**0.55**	***p* < 0.001**	**0.42–0.73**
Screen time 1 h before bed (0 = never/sometimes, 1 = often/always)						
≤1 sleep problem	1.00			1.00		
2 sleep problems	**1.69**	***p* < 0.001**	**1.36–2.10**	**1.61**	***p* < 0.001**	**1.30–2.00**
≥3 sleep problems	**2.78**	***p* < 0.001**	**2.12–3.64**	**2.59**	***p* < 0.001**	**1.98–3.38**
Screen time in bed (0 = never/sometimes, 1 = often/always)						
≤1 sleep problem	1.00			1.00		
2 sleep problems	**2.01**	***p* < 0.001**	**1.50–2.68**	**1.91**	***p* < 0.001**	**1.42–2.58**
≥3 sleep problems	**3.49**	***p* < 0.001**	**2.45–4.98**	**3.26**	***p* < 0.001**	**2.27–4.67**
Fruit consumption guidelines ^4^ (0 = not met, 1 = met)						
≤1 sleep problem	1.00			1.00		
2 sleep problems	0.84	*p* = 0.119	0.68–1.04	0.86	*p* = 0.175	0.70–1.07
≥3 sleep problems	**0.76**	***p* = 0.035**	**0.59–0.98**	0.79	*p* = 0.084	0.60–1.03
Vegetable consumption guidelines ^5^ (0 = not met, 1 = met)						
≤1 sleep problem	1.00			1.00		
2 sleep problems	**1.30**	***p* = 0.030**	**1.03–1.65**	**1.38**	***p* = 0.007**	**1.09–1.74**
≥3 sleep problems	1.06	*p* = 0.755	0.72–1.56	1.16	*p* = 0.450	0.79–1.70
Sugar sweetened beverage consumption (0 = ≤1/day, 1 = ≥2/day)						
≤1 sleep problem	1.00			1.00		
2 sleep problems	**1.81**	***p* = 0.001**	**1.26–2.60**	**1.70**	***p* = 0.004**	**1.19–2.43**
≥3 sleep problems	**2.24**	***p* < 0.001**	**1.60–3.16**	**1.96**	***p* < 0.001**	**1.37–2.80**
Sugar sweetened beverage 1 h before bed (0 = never/sometimes, 1 = often/always)						
≤1 sleep problem	1.00			1.00		
2 sleep problems	**1.86**	***p* = 0.004**	**1.21–2.84**	**1.82**	***p* = 0.007**	**1.18–2.80**
≥3 sleep problems	**2.37**	***p* = 0.001**	**1.43–3.91**	**2.21**	***p* = 0.003**	**1.32–3.72**
Takeaway consumption (0 = ≤1/week, 1 = ≥2/week)						
≤1 sleep problem	1.00			1.00		
2 sleep problems	**1.50**	***p* = 0.007**	**1.12–2.02**	**1.48**	***p* = 0.013**	**1.09–2.02**
≥3 sleep problems	1.35	*p* = 0.090	0.95–1.92	1.24	*p* = 0.240	0.87–1.77
Snack consumption (0 = ≤1/day, 1 = ≥2/day)						
≤1 sleep problem	1.00			1.00		
2 sleep problems	**1.45**	***p* = 0.001**	**1.17–1.79**	**1.42**	***p* = 0.001**	**1.15–1.77**
≥3 sleep problems	**1.32**	***p* = 0.047**	**1.00–1.74**	1.31	*p* = 0.057	0.99–1.72

# <1 sleep problems referent category; ^1^ adjusted for: school, study sample, gender, age, BMI-z, ICSEA (Index of Community Socio-educational Advantage), school clustering; ^2^ PA guidelines met ≥1 h/day on five or more days; ^3^ ST guidelines met ≤2 h/day on five or more days; ^4^ consumed ≥2 servings of fruit per day; ^5^ consumed ≥5.5 servings of vegetables for boys aged ≥12 years, or ≥5.5 servings for younger boys and all girls. Bold numbers indicate *p*-values < 0.05 or < 0.001.

**Table 4 ijerph-16-01778-t004:** Unadjusted and adjusted logistic regression with odds ratio (OR) for overweight/obesity with sleep score and behavioral factors.

Variables	Model 1			Model 2		
	OR	*p*	95% CI	OR	*p*	95% CI
Sleep Problems						
≤1	1.00			1.00		
2	**1.48**	***p* < 0.001**	**1.24–1.77**	**1.42**	***p* < 0.001**	**1.18–1.72**
≥3	**1.48**	***p* = 0.008**	**1.11–1.97**	**1.38**	***p* = 0.037**	**1.02–1.88**
Study Sample	1.01	*p* = 0.938	0.82–1.24	1.01	*p* = 0.921	0.82–1.25
Age	0.94	*p* = 0.076	0.87–1.01	0.95	*p* = 0.228	0.89–1.03
Gender	1.12	*p* = 0.195	0.94–1.34	1.06	*p* = 0.524	0.88–1.29
ICSEA	1.00	*p* = 0.050	0.99–1.00	1.00	*p* = 0.057	1.00–1.00
Physical activity guidelines ^†^						
Not met on 5 out of 7 days				1.00		
Met on 5 out of 7 days				**0.72**	***p* < 0.001**	**0.60–0.86**
Physical activity 1 h before bed						
Never/sometimes				1.00		
Often/almost always				**0.64**	***p* < 0.001**	**0.50–0.81**
Screen time guidelines ^#^						
Not met on 5 out of 7 days				1.00		
Met on 5 out of 7 days				0.87	*p* = 0.201	0.71–1.08
Screen time 1 h before bed						
Never/sometimes				1.00		
Often/almost always				1.12	*p* = 0.280	0.91–1.39
Screen time in bed						
Never/sometimes				1.00		
Often/almost always				1.02	*p* = 0.908	0.73–1.42
Sugar sweetened beverage consumption						
Once a day or less				1.00		
Twice or more per day				1.22	*p* = 0.197	0.90–1.65
Sugar sweetened beverage 1 h before bed						
Never/sometimes				1.00		
Often/almost always				0.89	*p* = 0.569	0.61–1.31

Model 1: DV ow/ob, IV sleep problems and activity factors controlled for demographic factors including study sample, age, Index of Community Socio-Educational Advantage (ICSEA), school clustering. Model 2 DV BMI-z, IV sleep problems and behavioral factors significantly associated with ow/ob or sleep. ^†^ PA guidelines met ≥1 h/day on five or more days; # ST guidelines met ≤2 h/day on five or more days. Bold numbers indicate *p*-values < 0.05 or < 0.001.

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
