# Peer review of "Dietary and Activity Factors Influence Poor Sleep and the Sleep-Obesity Nexus among Children"

_ijerph, 2019, doi:10.3390/ijerph16101778_

Reviewer 1 Report

This is a well-designed study, with an appropriate method and data analysis. The results are very interesting. I only suggest the following minor modifications:

Introduction >> Line 3: Finnish children instead of finish children  

Table 1: Under the sleep-wake categories, what do EL, EE, LL and LE stand for? Please, could you define those acronyms in the footnote.

Discussion >> Lines 295-297. According to the authors, screen devices and SSB consumption significantly increase the likelihood of having =3 sleep problems. I would be a little more cautious presenting these results, saying for example that those variables were associated with grater odds of having =3 sleep problems. Based on the empirical results, association is all that this study can prove and, so, the way it’s currently written suggests some kind of causal-effect relationship.

Discussion >> Lines 295-297. I’d state as a limitation the fact that the study sample does not represent the population of 10 years-old in Victoria and Australia. This brings some issues with the feasibility of extrapolating the results, which must be admitted.

Discussion >> I missed a paragraph stating implications for health and education policymaking.

Author Response

The authors would like to thank you for your positive and constructive comments. 

Please find attached a documents of responses to each of your comments. 

Thank you for your valued time in reviewing our manuscript.

Response to Reviewer 1 Comments

Comment   From Reviewer 1

Response:

Point 1:

Introduction   >> Line 3: Finnish children instead of finish children 

This   has been corrected.

(See highlighted change on line 72)

Point 2:

Table   1: Under the sleep-wake categories, what do EL, EE, LL and LE stand for?   Please, could you define those acronyms in the footnote.

These   have been added to the legend in Table 1 as below:

EL=   early to bed/late to rise; EE=   early to bed/late to rise; LL=late   to bed/late to rise; LE=late to   bed/late to rise

Point 3:

Discussion   >> Lines 295-297. According to the authors, screen devices and SSB   consumption significantly increase the likelihood of having =3 sleep problems.   I would be a little more cautious presenting these results, saying for   example that those variables were associated with grater odds of having =3   sleep problems. Based on the empirical results, association is all that this   study can prove and, so, the way it’s currently written suggests some kind of   causal-effect relationship.

The   wording has been changed to infer the association to be of increased odds   rather than causation.

(See highlighted changes in line   299 and below)

“were   shown to be associated with   increased odds of having ≥3 sleep problems”

Point 4:

Discussion   >> Lines 295-297. I’d state as a limitation the fact that the study   sample does not represent the population of 10 years-old in Victoria and   Australia. This brings some issues with the feasibility of extrapolating the   results, which must be admitted.

In   response to the study population limitations:

Text   has been added to further clarify the limitation of generalising results   beyond the current sample (Please see   highlighted changes in lines 412-415) and below.

“As mentioned, the current study   benefits from a large representative sample of children within two rural   regions of Victoria, with high school (GSC=68%, GV60%) and student (GSC=90%,   GV=78%) response rates.  However, although the current   population groups did not differ significantly, it cannot be assumed that the   reported associations can be generalised beyond children aged 8-13 years   within rural Victoria, Australia.”

Point 5:

Discussion   >> I missed a paragraph stating implications for health and education   policymaking.

Thank   you for this comment and our clear omission of this critical implications   piece.  The below text has been   added  (Please see highlighted changes in lines 400-408).

These findings support the importance of adhering to the 24-hour   movement guidelines outlined by the Australian Government Department of   Health, considering the role of multiple behaviours on the healthy   development and obesity risk among children and adolescents [1]. However, with current   results suggesting the importance sugar sweetened beverage consumption on children’s   sleep behaviours and weight status outcomes, there is a need for more   specific guidelines on these behaviours. Current Australian guidelines   recommend the intake of SSB be limited [2], however it would be   recommended the Government work with researchers to provide quantifiable   guidelines to limit this behaviour (similar to the recommended 2 hour screen   time limit).

References:

1.         Department of Health. Australian 24-Hour Movement Guidelines for Children and Young People (5-17 years) – An Integration of Physical Activity, Sedentary Behaviour and Sleep. Availabe online: http://www.health.gov.au/internet/main/publishing.nsf/Content/health-24-hours-phys-act-guidelines (accessed on 1st May 2019).

2.         National Health and Medical Research Council. Australian Dietary Guidelines. Availabe online: https://www.eatforhealth.gov.au/sites/default/files/content/The%20Guidelines/n55f_children_brochure.pdf (accessed on 21 Feb 2019).

Reviewer 2 Report

This is a large epidemiological study on diet, activity and sleep in children. The theme is interesting. I have some comments to improve the manuscript.

Major comments:

Authors think too much of statistical significance (P<0.05 and/or P<0.01). Recently, it was recommended not to mention statistical significance, but to present confidence intervals (https://www.nature.com/magazine-assets/d41586-019-00857-9/d41586-019-00857-9.pdf).

Please present participation rate of Great South Coast Childhood Obesity Monitoring Study and the Goulburn Valley Health Behaviours Monitoring Study, especially in the grade four and grade six. This information is important to know how the results represent the target populations.

Personally, I have favorable impression on your presenting all the regression analysis. However, there may be readers who concern multiple analysis. You can present your main outcomes in tables 3 and 4, and you can move the others to supplement. And if you present exact p-values, readers can perform Bonferroni correction to deal with multiple analysis.

In the discussion, please compare your study population with other studies analyzing the similar age range.

Physical activity and sleep are not recorded or monitored with devices. Study population is in one region in one country. And it is not clear that the results are also true in other locations. Please add these as study limitations.

Minor comments:

Lines 199-200: “serves”. Please clarify serves per day or per week.

Line 205: Please explain “takeaway” more. For non-natives, it is not easy to understand “takeaway”.

Table 1: Please spell out ICSEA, EL, EE, LL and LE in the legend.

Throughout the tables: please present exact p-values instead of P<0.01, p<0.05 and NS (for example, P=0.352, P=0.024, P=0,003 or P<0.001).< p="">

Table 2: please add total number of subjects who had one or less, two, three or more sleep dimensions scored poor at the top of the table.

Author Response

The authors would like to thank you for your positive and constructive comments. 

Please find attached a document of our responses to each of your comments. 

Thank you for your valued time in reviewing our manuscript.

Response to Reviewer 2 Comments

Comment   From Reviewer 2

Response:

Major   comments:

Point 1:

Authors   think too much of statistical significance (P<0.05 and/or P<0.01).   Recently, it was recommended not to mention statistical significance, but to   present confidence intervals   (https://www.nature.com/magazine-assets/d41586-019-00857-9/d41586-019-00857-9.pdf).

The   authors have removed the over emphasis on statistical significance, firstly   by removing language indicating the existence or lack of a significant   difference or association. Secondly, taking into account your later comment   on reporting exact p-values, these have now been provided.  Where relevant the confidence intervals   (previously provided) are also reported. This will allow readers to interpret   the occurrence of associations for themselves. 

Point 2:

Please   present participation rate of Great South Coast Childhood Obesity Monitoring   Study and the Goulburn Valley Health Behaviours Monitoring Study, especially   in the grade four and grade six. This information is important to know how   the results represent the target populations.

Data   around school and student response rates have been included in the methods   section, including a breakdown per region.

(Please see highlighted changes   in lines 106-114)

Point 3:

Personally,   I have favorable impression on your presenting all the regression analysis.   However, there may be readers who concern multiple analysis. You can present   your main outcomes in tables 3 and 4, and you can move the others to   supplement. And if you present exact p-values, readers can perform Bonferroni   correction to deal with multiple analysis.

The   authors agree with being in favour of presenting all regression analysis   tables, and have maintained these within the manuscript.

As   previously mentioned, the authors have also provided the exact p-values for   the readers benefit.

Point 4:

In   the discussion, please compare your study population with other studies analyzing   the similar age range.

Where   possible studies of similar samples were used to discuss previous finding   with those of the current study (age ranges 8-16) [1-8]. However, due to   limited available studies focussing on the different dimensions of sleep and   the multiple behaviours of physical activity, screen-time and diet, studies   including slightly older adolescent populations (age ranges 6-19) [9-13] and adults [14,15] have been   incorporated to further unpack the findings of the current sample.

Throughout   the discussion text has been added to more clearly highlight the age group of   studies used for comparison.

Point 5:

Physical   activity and sleep are not recorded or monitored with devices. Study   population is in one region in one country. And it is not clear that the   results are also true in other locations. Please add these as study   limitations.

In   response to the study population limitations:

Text   has been added to further clarify the limitation of generalising results   beyond the current sample  as   highlighted by Reviewer 1. (Please see   highlighted changes in lines 412-415).

“As mentioned, the current study   benefits from a large representative sample of children within two rural   regions of Victoria, with high school (GSC=68%, GV60%) and student (GSC=90%,   GV=78%) response rates.  However, although the current   population groups did not differ significantly, it cannot be assumed that the   reported associations can be generalised beyond children aged 8-13 years   within rural Victoria, Australia.”

In   response to the limitations around the measurements used for physical   activity and sleep:

This   had been mentioned in the limitations initially. Text has been added to   further emphasise the need for more objective measures. (Please see highlighted changes in lines 416-422 and below).

“Furthermore, while questionnaire   items were sourced from studies with previous validated question items   [48-50] with these and similar self-report questionnaires validated among   young children, the self-report nature of the sleep and behavioural factors   items could be subject to recall or report bias [79]. While there is evidence supporting the   suitability for use of questionnaires in the examination of the association   with weight status [80], objective measures (such as accelerometry) are   deemed more accurate for assessment of sleep and physical activity behaviours.”  

Minor   comments:

Point 6:

Lines   199-200: “serves”. Please clarify serves per day or per week.

This   has been amended.

(See highlighted changes on lines   199-201)

Point 7:

Line   205: Please explain “takeaway” more. For non-natives, it is not easy to   understand “takeaway”.

This   has been amended as below.

(See below and highlighted change   on line 205)

“(meals purchased from restaurants or fast-food vendors)”

Point 8:

Table   1: Please spell out ICSEA, EL, EE, LL and LE in the legend.

Thank   you for this comment, these have been added to the legend in Table 1 as   below:

ICSEA= Index of Community Socio-Educational   Advantage; EL= early to bed/late to rise; EE= early to bed/late to rise; LL=late to bed/late to rise; LE=late to bed/late to rise

Point 9:

Throughout   the tables: please present exact p-values instead of P<0.01, p<0.05 and   NS (for example, P=0.352, P=0.024, P=0,003 or P<0.001).

Thank  you for this comment, as mentioned above   (Point 1), this has been addressed throughout.

Point 10:

Table   2: please add total number of subjects who had one or less, two, three or   more sleep dimensions scored poor at the top of the table.

This   had initially been left out as these figures are outlined in Table One, as   well as to minimise clutter.

This   has now been amended.

(See highlighted changes at the   top of Table 2)

References:

1.         Harrex, H.A.L.; Skeaff, S.A.; Black, K.E.; Davison, B.K.; Haszard, J.J.; MeredithJones, K.; Quigg, R.; Saeedi, P.; Stoner, L.; Wong, J.E., et al. Sleep timing is associated with diet and physical activity levels in 911yearold children from dunedin, new zealand: The pedals study. Journal of sleep research 2017, 10.1111/jsr.12634, doi:10.1111/jsr.12634.

2.         Khan, M.K.A.; Yen Li, C.; Kirk, S.F.L.; Veugelers, P.J.; Chu, Y.L. Are sleep duration and sleep quality associated with diet quality, physical activity, and body weight status? A population-based study of Canadian children. Canadian Journal of Public Health 2015, 106, e277-e282, doi:10.17269/cjph.106.4892.

3.         Sampasa-Kanyinga, H.; Hamilton, H.A.; Chaput, J.P. Sleep duration and consumption of sugar-sweetened beverages and energy drinks among adolescents. Nutrition 2018, 48, 77-81, doi:10.1016/j.nut.2017.11.013.

4.         Stone, M.R.; Stevens, D.; Faulkner, G.E.J. Maintaining recommended sleep throughout the week is associated with increased physical activity in children. Preventive Medicine: An International Journal Devoted to Practice and Theory 2013, 56, 112-117, doi:10.1016/j.ypmed.2012.11.015.

5.         Chahal, H.; Fung, C.; Kuhle, S.; Veugelers, P.J. Availability and night-time use of electronic entertainment and communication devices are associated with short sleep duration and obesity among Canadian children. Pediatric Obesity 2013, 8, 42-51.

6.         de Jong, E.; Stocks, T.; Visscher, T.L.S.; HiraSing, R.A.; Seidell, J.C.; Renders, C.M. Association between sleep duration and overweight: the importance of parenting. International Journal of Obesity 2012, 36, 1278-1284, doi:10.1038/ijo.2012.119.

7.         McNeil, J.; Tremblay, M.S.; Leduc, G.; Boyer, C.; Belanger, P.; Leblanc, A.G.; Borghese, M.M.; Chaput, J.P. Objectively-measured sleep and its association with adiposity and physical activity in a sample of Canadian children. Journal of sleep research 2014, 10.1111/jsr.12241, doi:10.1111/jsr.12241.

8.         Wells, J.C.K.; Hallal, P.C.; Reichert, F.F.; Menezes, A.M.B.; Araújo, C.L.P.; Victora, C.G. Sleep patterns and television viewing in relation to obesity and blood pressure: evidence from an adolescent Brazilian birth cohort. International Journal of Obesity 2008, 32, 1042-1049, doi:10.1038/ijo.2008.37.

9.         Golley, R.K.; Maher, C.A.; Matricciani, L.; Olds, T.S. Sleep duration or bedtime? Exploring the association between sleep timing behaviour, diet and BMI in children and adolescents. International Journal of Obesity 2013, 37, 546-551.

10.       Gupta, N.K.; Mueller, W.H.; Chan, W.; Meininger, J.C. Is obesity associated with poor sleep quality in adolescents? American Journal of Human Biology 2002, 14, 762-768, doi:10.1002/ajhb.10093.

11.       Hitze, B.; Bosy-Westphal, A.; Bielfeldt, F.; Settler, U.; Plachta-Danielzik, S.; Pfeuffer, M.; Schrezenmeir, J.; Mönig, H.; Müller, M.J. Determinants and impact of sleep duration in children and adolescents: data of the Kiel Obesity Prevention Study. European Journal Of Clinical Nutrition 2009, 63, 739-746, doi:10.1038/ejcn.2008.41.

12.       Beebe, D.W.d.b.c.o.; Lewin, D.; Zeller, M.; McCabe, M.; MacLleod, K.; Daniels, S.R.; Amin, R. Sleep in Overweight Adolescents: Shorter Sleep, Poorer Sleep Quality, Sleepiness, and Sleep-Disordered Breathing. Journal of pediatric psychology 2007, 32, 69-79, doi:10.1093/jpepsy/jsj104.

13.       Fleig, D.; Randler, C. Association between chronotype and diet in adolescents based on food logs. Eat Behav 2009, 10, 115-118, doi:10.1016/j.eatbeh.2009.03.002.

14.       St-Onge, M.P.; Roberts, A.; Shechter, A.; Choudhury, A.R. Fiber and Saturated Fat Are Associated with Sleep Arousals and Slow Wave Sleep. J Clin Sleep Med 2016, 12, 19-24, doi:10.5664/jcsm.5384.

15.       Vallance, J.K.; Buman, M.P.; Stevinson, C.; Lynch, B.M. Associations of overall sedentary time and screen time with sleep outcomes. American journal of health behavior 2015, 39, 62-67.

Round  2

Reviewer 2 Report

Authors fairly well revised the manuscript.

I have only one minor comment.

Please move the explanation of takeaway from line 205 to line 174 (first appear here).

Author Response

Revised.